# MicroRNAs as Potential Biomarkers for Exercise-Based Cancer Rehabilitation in Cancer Survivors

**DOI:** 10.3390/life11121439

**Published:** 2021-12-20

**Authors:** Yanping Jiang, Kulsoom Ghias, Sanjeev Gupta, Ananya Gupta

**Affiliations:** Department of Physiology, National University of Ireland, H91 TK33 Galway, Ireland; Y.Jiang7@nuigalway.ie (Y.J.); k.ghias1@nuigalway.ie (K.G.); sanjeev.gupta@nuigalway.ie (S.G.)

**Keywords:** miRNA, cancer survivorship care, rehabilitation, personalized exercise intervention, molecular biomarker

## Abstract

Expression and functions of microRNAs (miRNAs) have been widely investigated in cancer treatment-induced complications and as a response to physical activity, respectively, but few studies focus on the application of miRNAs as biomarkers in exercise-based cancer rehabilitation. Research has shown that certain miRNA expression is altered substantially due to tissue damage caused by cancer treatment and chronic inflammation. MiRNAs are released from the damaged tissue and can be easily detected in blood plasma. Levels of the miRNA present in peripheral circulation can therefore be used to measure the extent of tissue damage. Moreover, damage to tissues such as cardiac and skeletal muscle significantly affects the individual’s health-related fitness, which can be determined using physiologic functional assessments. These physiologic parameters are a measure of tissue health and function and can therefore be correlated with the levels of circulating miRNAs. In this paper, we reviewed miRNAs whose expression is altered during cancer treatment and may correlate to physiological, physical, and psychological changes that significantly impact the quality of life of cancer survivors and their role in response to physical activity. We aim to identify potential miRNAs that can not only be used for monitoring changes that occur in health-related fitness during cancer treatment but can also be used to evaluate response to exercise-based rehabilitation and monitor individual progress through the rehabilitation programme.

## 1. Rehabilitation in Cancer Survivors

It is estimated that there were 18.1 million newly diagnosed cancer cases and 9.6 million cancer-related deaths globally in 2018 [1]. Despite the advances in cancer treatment, cancer incidence and mortality have been increasing over the years, thus making cancer one of the greatest health threats to human beings [2]. Cancer treatment strategies, such as surgery, chemotherapy, and radiotherapy, increase survival and prolong the lifespan of cancer survivors. However, these strategies have many adverse effects on patients’ health-related fitness (including cardiovascular endurance, muscular strength, muscular endurance, flexibility, body composition). Evidence suggests patients recovering from cancer have an increased risk of developing myriad chronic, comorbid conditions, such as metabolic syndrome, fatigue, muscle wasting (sarcopenia/cachexia), bone loss, lymphoedema, cardiotoxicity, and mental health issues, and respond poorly to standard treatment, which makes it challenging to treat such conditions [3,4,5,6,7]. The risk of developing long-term side effects following cancer treatment is influenced by: age at diagnosis; comorbidities; cancer type and stage; and treatment type, dose, and duration [7]. Additionally, these side effects not only occur during treatment but can emerge many years after the completion of treatment. This, in turn, leads to a long-term illness burden and morbidity and an increased risk of premature mortality [4,5]. Therefore, the importance of addressing the long-term needs of cancer survivors is well-recognised [8].

According to the National Cancer Registry Ireland (NCRI), 22,640 patients are newly diagnosed with cancer every year in Ireland [4]. Cancer survivorship begins at the time of diagnosis and continues until the end of life and is broadly referred to as ‘*living with and beyond cancer’* [4]. With improvements in cancer treatment, the survival rate has improved significantly, and, for most cancers, is estimated to be 77–98% at one year and 63–92% at five years posttreatment. There were 173,000 cancer survivors living in Ireland in 2016, and this number is predicted to exceed 200,000 in 2020 [5]. The National Cancer Strategy Ireland 2017–2026 [4] outlined the need to establish support services providing effective management of posttreatment health issues to improve the quality of life (QoL) of cancer survivors. The development of such a service has been hindered by a lack of research and information defining the posttreatment needs of cancer patients. In this regard, great efforts should be directed to develop rehabilitation strategies based on cancer survivors’ needs to reduce or alleviate cancer treatment-induced complications.

Exercise-based cancer rehabilitation is one of these strategies and has been proved to be an effective way to improve the QoL in cancer survivors. There are three main types of exercise training included as part of cancer rehabilitation: resistance exercise, aerobic exercise, and the combination of both [9]. Flexibility and mobility exercises aiming to restore the normal range of motion also plays an integral part in cancer rehabilitation. Recent studies have overwhelmingly supported that exercise-based cancer rehabilitation programs can alleviate, prevent, and even treat the side effects caused by cancer treatment. A randomized controlled trial from overweight or obese breast cancer survivors showed that, when compared with usual care, participants who received 16 weeks of combined aerobic and resistance exercise intervention had physiologically improved metabolic syndrome z-score, sarcopenic obesity, and circulating biomarkers, such as insulin, insulin-like growth factor-1 (IGF-1), leptin, and adiponectin [10]. In non-small cell lung cancer patients, after initial surgical resection, 8-week combined aerobic exercise and high-intensity respiratory muscle training can significantly improve peak oxygen uptake (VO_2_peak), exercise capacity, and respiratory muscle strength [11]. Breast cancer patients who received adjuvant chemotherapy were more susceptible to develop sarcopenia and dynapenia. While these symptoms can be reversed by resistance exercise training undertaken during and after treatment, such reversal was correlated to a significant improvement of the long-term outcome of breast cancer survivors [12]. In a systematic review of 16 trials with cancer patients undergoing treatment, physical exercise was reported to generate better effects than usual care, especially on maintaining muscle mass and strength [9]. Despite not very strong evidence, resistance exercise seems to be more effective than aerobic exercise both on muscle mass and strength, which was consistent with the findings of Courneya et al. [13]. In addition, cancer patients who received exercise intervention had reduced depressive symptoms and improved emotional functions as well as QoL compared to their counterparts [11,14,15,16]. Taken together, exercise-based cancer rehabilitation contributes significantly to the improvement of the QoL of cancer survivors by improving health-related fitness and physical function.

In addition to a supervised exercise programme, home-based exercise also showed promising prospects in cancer rehabilitation. A multicentre randomized controlled trial suggested that home-based physical activity consisted of combined endurance (same as aerobic exercise) and resistance training during 27 weeks significantly increased VO_2_peak compared to a decrease in the usual care group [17]. In addition to endurance and resistance exercise, home-based walking training was also reported to benefit breast cancer patients for cardiorespiratory fitness [18]. These results aligned with the hospital-based rehabilitation programs, revealing that home-based physical activity was favourable to cancer patients under adjuvant or neoadjuvant chemotherapy [19,20]. Moreover, this kind of home-based rehabilitation program was of great feasibility and practicality if patients were well instructed. In summary, exercise-based cancer rehabilitation is beneficial to cancer survivors regardless of the type and place. Therefore, a properly structured exercise-based rehabilitation programme is an unmet need for cancer patients undergoing treatment.

## 2. Introduction to miRNAs

miRNAs are short, 20-22-nucleotide, noncoding RNAs that inhibit the expression of their cognate target genes by directly binding to their mRNAs, thus blocking their translation and targeting them for degradation. In this way, miRNAs play an essential part in regulating gene expression in cells. Various studies have elucidated that the aberrant expression of miRNAs is associated with the onset and progress of multiple diseases, including cancer, diabetes, inflammation, cardiovascular disease, neurological disorders, etc. Evidence has also shown that miRNAs can act as intracellular modulators of responses to cancer treatment and treatment-induced complications or toxicities [21,22]. MiRNAs have been attributed with several characteristics such as (1) presence in multiple body fluids, including blood; (2) high stability; (3) resistance to RNA enzyme, thus resulting in long half-life; (4) tissue-specific function; and (5) easy detection and quantitation by different methods. These properties make miRNAs amenable to a wide range of applications in diagnosis, prognosis, and treatment in multiple disease settings.

Cells constantly secrete small vesicles (30–150 nm) called exosomes, which are found in abundance in all body fluids including blood. These vesicles contain RNA and protein secreted from the cells from which the exosome originates and are involved in many biological functions, including cell-to-cell communication and signalling [23,24]. Since exosomes represent a snapshot of the cellular contents, any condition that affects the normal functions of the cell is likely to alter the composition of exosomes. As a result, RNA and protein present in the exosomes can be used as biomarkers that can have both prognostic and diagnostic applications [23,24]. Consequently, profiling of circulating RNAs, in particular miRNAs, have been used in a number of studies to identify novel and highly promising biomarkers for many pathologies including neurodegenerative diseases such as Alzheimer’s and Parkinson’s, brain injury, presence of brain tumours, and in cancer (breast and pancreatic cancers) [23,24]. Many of these molecules were determined to be associated with exosomes. Exosomes can be easily purified and analysed and therefore have gained importance in research as a source for identification of novel biomarkers. For instance, miRNAs in exosomes are reported to be promising biomarkers in Alzheimer’s screening and progression monitoring as well as chemoresistance and metastasis in cancer [23,24]; miR-375 and miR-9 have shown promise as prognostic biomarkers for prediabetes and Type 2 Diabetes Mellitus (T2DM) [25]; antagomiRs, the synthetic antagonists of miRNAs, could compete with the target mRNA for binding with its cognate miRNA, thus inhibiting miRNA function. This helps to inhibit miRNA-mediated suppression of gene expression and seems encouraging in various diseases as a therapeutic approach as inhibitor of miRNAs [26]; examples are shown in Table 1. 

Aberrant miRNA expression, either downregulated or upregulated in various cancer cell lines or clinical tumour tissues, can not only directly reflect chromosomal or genomic changes in cancer-related genes but also appear to have clinical significance. For example, the change in expression of these molecules has been linked to disease severity and prognosis [35]. Changes in the expression and function of certain of these miRNAs have also been related to the course of disease [35]. The researchers found that circulating miRNA levels can be used to identify serum samples from cancer patients from samples from healthy people and samples taken before and after surgery by analysing specific miRNAs in serum or blood plasma. Heneghan and colleagues found elevated miR-195 expression in 148 BC cancer patients using real-time PCR analysis. Furthermore, two weeks following breast reduction, circulating miR-195 and let-7a levels were shown to decrease in patients’ blood, reaching the expression level seen in the healthy group [31]. Schooneveld and colleagues found substantial (*p* < 0.05) variations in miR-215, miR-299-5p, and miR-411 expression levels in sera samples from 20 healthy controls and 75 patients with BC, including 16 patients with untreated metastatic breast. The study revealed that miRNA expression was downregulated in malignant serum samples; however, the expression levels reverted to normal eight weeks after surgical treatment [36]. In another investigation, Enders et al. [32] found a statistically significant (*p* < 0.005) drop in plasma levels of miRNAs in 170 BC patients. The authors discovered a combination plasma biomarker of miR-451 and miR-145 that is highly effective in identifying BC patients from healthy volunteers (*p* < 0.0001). As a result, miRNAs have become a potential noninvasive tool for cancer diagnosis and prognosis.

MiRNA expression and function has also been investigated in exercise and fitness, especially in the exercise physiology of athletes [37]. For instance, miRNA is regarded as a key regulator of the signalling pathways of exercise adaptation, such as the IGF1/PI3K/AKT/mTOR axis, and is associated with phenotypic features, such as VO_2_max. Therefore, miRNA expression patterns were proposed to define responses to exercise and could also be utilized to assess physical performance and capacity through monitoring changes in physiologic parameters that determine health-related fitness [38]. Davidsen et al. [39] found miR-378, miR-29a, and miR-26a were downregulated in low responders to resistance exercise and unchanged in high responders, whereas miR-451 was upregulated in high responders, suggesting that the specific expression pattern of miRNAs may help evaluate the responsiveness to physical activity. Domańska-Senderowska et al. [40] and Polakovičová et al. [41] also reviewed miRNAs and their roles in physical activity. In addition, due to its role in the cardiorespiratory and muscular system, miRNA has also been investigated in the rehabilitation program. For example, downregulated has-miR-125b-1-3p and upregulated has-miR-1290 were found in heart failure patients who received cardiac rehabilitation (2 weeks of bicycle ergometer and resistance exercises) [42]; miR-335-3p and miR-657, posttranscriptional regulators of IL-37 production, were increased after a rehabilitation program in sarcopenia patients, accompanied by improvement of physical and cognitive conditions [43]. These works have shown that miRNAs may be promising biomarkers for a rehabilitation program.

Although miRNAs have been widely studied in diseases and in response to physical activity (as shown in Figure 1), few studies have investigated the application of miRNAs as a dynamic monitoring tool during exercise-based cancer rehabilitation. In the following sections, we will review the relationship between miRNAs and common treatment-induced complications, namely (1) metabolic syndrome, (2) cardiorespiratory toxicity, (3) cancer cachexia, and (4) depression and anxiety. We also summarize current knowledge on how physical activity can influence these miRNAs, particularly in the exercise-based cancer rehabilitation program. Finally, we will discuss how these miRNAs can be used to predict the QoL of cancer survivors. We aim to identify potential miRNAs for future investigation to evaluate or monitor progress in exercise-based rehabilitation for cancer survivors.

## 3. miRNAs in Cancer Treatment-Induced Complications and Physical Activity

### 3.1. Metabolic Syndrome

Metabolic syndrome is a clustering of metabolic disorders, including insulin resistance, dyslipidaemia, and high blood pressure, which can contribute to T2DM and cardiovascular disease (CVD) [44]. Available evidence indicates that metabolic syndrome may increase the risk of common solid tumours, although the risk may differ between the sexes and ethnicity [45]. Moreover, the metabolic syndrome might be the causative link between cancer treatment and cardiac dysfunction in cancer survivors, as it is associated with pre-inflammatory, inflammatory, and prethrombotic effects. These responses worsen cancer-related comorbidity and mortality [46,47]. Cancer treatment itself can also lead to the development of the metabolic syndrome. Studies have shown that the risk of suffering treatment-related metabolic syndrome in cancer patients was 1.31- to 4.58-fold greater than in the control group [48,49]. Currently, three main factors contribute to cancer treatment-induced metabolic syndrome: (1) inhibition or damage to the pituitary and hypothalamus; (2) resection or toxicity to gonadal glands and other endocrine glands; and (3) physical inactivity. These factors result in chronic hormonal disturbance and calorie imbalance, which in turn lead to obesity, insulin resistance, impaired lipid transport, impaired glucose uptake, and, eventually, to increased risk of CVD [46].

Research has shown that miRNAs are key regulators of metabolic syndrome in cancer patients. miRNA-mediated glucose metabolism and lipid metabolism dysregulation are the main manifestations of cancer treatment-induced metabolic syndrome. These miRNAs are predominantly expressed in macrophages and adipocytes and subsequently released into the systemic circulation. These miRNAs are taken up by recipient cells, such as adipocytes and hepatocytes, in which the miRNAs regulate a series of cellular processes, such as insulin secretion, insulin sensitivity, pancreas homeostasis, lipid storage, and inflammation of adipose tissue (for review, see Zhang et al. [50]). A group of miRNAs were shown to play an essential role in metabolic syndrome (for review, see Huang et al. [51] and Eckel et al. [52]) through a mechanism involving the *PI3K/Akt* and *TGF-β/Smad3* signalling pathways as well as their target genes, such as *PTEN, GLUT4, IRS-1* [51,52].

As physical activity and exercise training can manage weight gain and balance calories, exercise is a promising treatment strategy for cancer treatment-induced metabolic syndrome. A population-based prospective cohort study recruiting 1696 breast cancer survivors showed that regular exercise diminished the prevalence of metabolic syndrome from 55.18% at baseline to 33.14% at 60 months postdiagnosis [53]. Zhou et al. [54] recruited 209 subjects with metabolic syndrome and 234 controls to investigate the correlation between physical activity-related miRNAs and metabolic syndrome. Results showed that, among 55 miRNAs detected, only miR-197, miR-126, and miR-130a were significantly correlated to physical activity. Moreover, the expression of miR-126 and miR-130a was lower in the most active participants with the highest metabolic equivalent hours per week of physical activity (MET-h/week). In other words, individuals with the highest MET-h/week had a lower risk of metabolic syndrome. By contrast, miR-197 expression was elevated with increased physical activity or MET-h/week. However, after adjustment for age, sex, and disease settings, results showed that miR-126 significantly increased the risk of metabolic syndrome. At the same time, miR-197 decreased the risk, suggesting the correlation between physical activity and metabolic syndrome risk may partly rely on miR-126 and miR-197.

### 3.2. Cardiorespiratory Toxicity

Anticancer treatment is usually accompanied by reduced cardiorespiratory function, which conversely leads to dose reduction or cessation of anticancer therapy. Cardiorespiratory toxicity in cancer results from chemotherapy-induced damage to cardiomyocytes, leading to cardiomyopathy, heart failure, acute myocardial infarction, arrhythmia, pulmonary fibrosis. The cardiorespiratory complication is one of the biggest causes of cancer-related mortality. These complications give rise to poor cardiac contractility and decreased ventricular ejection fraction, which causes insufficient oxygen transported by the circulatory system. The skeletal muscles need oxygen to generate adenosine triphosphate (ATP) to produce contraction; thus, poor oxygen-carrying capacity limits the individual physical capacity and ability to perform activities of daily living. This, therefore, dramatically reduces the QoL of cancer survivors [55]. Two major factors that contribute to reduced cardiovascular capacity induced by cancer treatment include (1) direct damage to cardiac muscle cells or lung tissue and (2) long-term cardiac disease caused by cancer treatment-induced metabolic syndrome. However, the exact mechanism of cardiotoxicity remains unknown. Evidence showed that miRNAs serve as key mediators in heart muscle cell proliferation, differentiation, and apoptosis. The most extensively investigated are the myomiR family, which refers to the miRNAs most expressed in the muscle tissue. Most myomiRs are expressed both in heart and skeletal muscle: miR-1, miR-133a, miR-133b, miR-486, and miR-499 [56], while some are only expressed in one muscle tissue: cardio-specific miR-208a and skeletal muscle-specific miR-206 [57]. Evidence illustrates that the dysregulation of myomiRs, along with metabolism-related miRNAs, was correlated to adverse cardiac remodelling and toxicity in both preclinical and clinical studies (for review, see Pellegrini et al. [22], Pereira et al. [58], and Riggeri et al. [59]). In neonatal cardiac rat models, the doxorubicin-induced upregulation of miR-146a facilitated cell death in myocytes by targeting ErbB4 and therefore caused cardiotoxicity [60]. The upregulation of miR-146a was also observed in doxorubicin-treated breast cancer patients [61]. Compared to non-anthracycline chemotherapy, the expression of miR-29a and miR-499 in plasma was elevated after anthracycline exposure. In addition, their higher expression in patients receiving anthracycline significantly correlated to dose and increased troponin levels [62].

Physical activity can improve cardiorespiratory function. Evidence showed that physiological response to exercise training results in the release of miRNAs in blood, which regulated the adaptive changes, such as muscle atrophy/hypertrophy, reflected by protein, glucose, and lipid metabolism, in the human body via IGF1/PI3K/AKT/mTOR signalling pathway. Therefore, great efforts are invested in exploring the effect of physical activity on miRNAs common to this particular pathway (for review, see Domańska-Senderowska et al. [40]). The expression of miRNA was significantly associated with changes in certain physiological parameters during exercise. The study showed that the expression of miR-1, -133a/b, and miR-486 was correlated to VO_2_max and anaerobic threshold [40], while circulating miR-486 was inversely correlated to resting heart rate [63]. These results indicated miRNAs might serve as biomarkers for cardiorespiratory function and fitness during physical activity.

### 3.3. Cancer Cachexia

Cancer cachexia is a complex multifactorial syndrome characterized by weight loss with the reduction of skeletal muscle and fat mass that affects 20% of cancer patients and is more prominent in refractory cancer patients [64]. The difference between malnutrition and cachexia is that cachexia is caused by loss of muscle mass and cannot be reversed by conventional nutritional interventions [65]. As a result, cachexia is deemed an independent predictor of cancer mortality. In animal models of cancer cachexia, the reversal of muscle loss gives rise to longer survival, making the biomarkers that regulate or mediate muscle atrophy to be of primary importance [66]. Recent work has shown that miRNAs were associated with the signalling pathway involved in the pathogenesis of cancer cachexia, especially the pathway of turnover of skeletal muscle and adipose tissue (for review, see Santos et al. [21]). Specifically, miRNAs modulate adipose and skeletal muscle tissue metabolism in cancer cachexia and also regulate tumour- and tissue-derived inflammation. Other studies have revealed that *PI3K/Akt/mTOR* and *Akt/FoxO* signalling pathways may play an important role in miRNA-mediated muscle atrophy. Narasimhan et al. [67] profiled the differentially expressed miRNAs in skeletal muscle between cachectic and non-cachectic cancer patients and identified eight upregulated miRNAs (upregulated, fold change of ≥1.4 at *p* < 0.05), namely hsa-miR-3184-3p, hsa-miR-423-5p, hsa-let-7d-3p, hsa-miR-1296-5p, hsa-miR-345-5p, hsa-miR-532-5p, hsa-miR-423-3p, and hsa-miR-199a-3p. In this work, Narasimhan et al. [67] also analysed the target gene and possible pathway to regulate cachexia including pathways involved in cytokine signalling and inflammation. Furthermore, these miRNAs showed significantly prognostic and predictive value in non-small cell lung cancer patients with cachexia, suggesting miRNAs may serve as potential biomarkers in cancer cachexia.

Physical activity is a potent nonpharmaceutical physiologic intervention to counteract cancer cachexia by improving and maintaining skeletal mass and strength. One of the possible mechanisms for this could be that physical activity brings about the changes of skeletal muscle-related miRNAs. MiRNAs play an intricate role in muscle biogenesis and development. Dysregulation of myomiRs, key mediators for muscle development, was associated with the dysfunction of the pathway that mediates myogenesis, muscle hypertrophy, and atrophy [68]. miRNAs were also associated with physiological measurements of cancer cachexia after exercise. Plasma miR-146a and miR-221 positively correlated with muscle mass and fat mass but negatively related to BMI [69]. Plasma miR-222 levels were positively associated with a strength-related performance measure; in contrast, plasma miR-21, miR-221, and miR-146a levels were negatively related to a subset of strength-related performance measures [69].

### 3.4. Depression and Anxiety

Depression and anxiety are two of the most common psychological symptoms in cancer patients. Cancer diagnosis, cancer treatment, and all the accompanying side effects that lead to reduced QoL may cause and even aggravate depression and anxiety. As a result, depression and anxiety are more severe in cancer survivors than in the general population. Evidence showed that the prevalence of depression of cancer patients varied among different cancer types, ranging from 1.5 to 58% [70]; anxiety ranged from 6 to 32.2% [71,72]. A study on depression in hospitalized cancer patients showed that 42% of oncologic inpatients met the criteria of major depression, with 24% with severe and 18% with moderately severe depression. In addition, 14% of cancer patients presented depressive symptoms though they did not meet the criteria of major depression [73]. Depression and anxiety have further negative effects on the QoL and correlated to elevated mortality in cancer survivors. Evidence supported that the mortality rate of cancer survivors who had depressive symptoms was 25%, but this rate was elevated to 39% when cancer patients were diagnosed with depression [74]. Mean survival at 24 months for cancer patients without depression and anxiety was 23.11 months versus 20.87 months for those with [75]. So far, the depression or anxiety self-rate scale is the most commonly used tool in diagnosing and measuring depression or anxiety. Therefore, it is necessary to develop biomarkers as an early screening and intervention strategy for depression and anxiety in cancer patients. The aberrant expression of noncoding transcriptomic substances, such as miRNA, long noncoding RNA, and messenger RNA, lead to the impaired functions of hippocampus, which is associated with a variety of mental health issues, including depression and anxiety, and thus serve as biomarkers in depressive symptoms [40]. In particular, miRNAs had been proved to play an important role in both clinical depression and in animal models (for review, see Gururajan et al. [76], Malan-Müller et al. [77]). Evidence showed that the aberrant expression of miR-17-92 clusters in hippocampal progenitor significantly influenced neurogenesis and anxiety- and depression-related behaviours: miR-17-92 knockout induced anxiety and depression; miR-17-92 overexpression exhibited antianxiety- and antidepression-like behaviour. This may be involved in neurogenesis-related genes, serum- and glucocorticoid-inducible protein kinase-1, in the glucocorticoid pathway [78]. Physical activity is an effective way to relieve depressive and anxious symptoms. In a study of effects of exercise dose and type on psychological distress in breast cancer patients, the recruited patients were assigned into three groups: a standard dose of aerobic exercise (25 to 30 min), a higher dose of aerobic exercise (50 to 60 min), and the combination of aerobic and resistance exercise (50 to 60 min). All these exercise types can alleviate depressive symptoms in patients with clinical levels of depression at baseline. Still, this function cannot be observed among unselected breast cancer patients undergoing chemotherapy [79]. Another physical exercise rehabilitation program (10-week twice-weekly) also reported the apparent improvement of depression and anxiety in breast cancer patients [80]. Mounting evidence showed that aerobic exercise could improve the function of the hippocampus, the mechanism of which involved SIRT1/miRNAs pathway, thus mitigating depressive symptoms [40,81]. However, few studies investigated the relationship between physical activity-related miRNAs and cancer treatment-related depression and anxiety. Hence, further studies are required to investigate the changing pattern of miRNAs in cancer-related depression and anxiety during exercise-based cancer rehabilitation.

We summarized the miRNA expression in common cancer treatment-induced complications and physical activity in Figure 2, aiming to look for the potential biomarkers for exercise-based cancer rehabilitation.

## 4. Potential miRNAs as Biomarkers in Exercise-Based Cancer Rehabilitation

### 4.1. Metabolic Syndrome

#### 4.1.1. miR-126 and miR-146a

Gomes et al. [82] showed that obesity could lead to decreased expression of miR-126 and an increased *PI3KR2*, which inhibits a key regulator of the *VEGF* signalling pathway *PI3K*, thus causing capillary rarefaction in skeletal muscle in the obese rat model, whereas aerobic exercise training, such as swimming, can reverse this negative impact by normalizing miR-126 level and restoring *VEGF* signalling. Of note, other studies showed miR-126 increased in obese patients, but after physical activity, there was a slight decrease of miR-126, without statistical significance [83].

miRNA was associated with the metabolic disorder by regulating the inflammatory response in white adipose tissue, leading to insulin resistance. miR-146a can suppress the macrophage-conditioned medium-induced inflammatory response in adipocytes, which was reflected by a decrease of *IL-8, MCP-1* mRNA and protein. The mechanism of this process involves a reduced inflammation-induced activation of *JNK* and *p38* via targeting *IRAK1* and *TRAF6* in human adipocytes [84]. Studies [83] showed that miR-146a-5p significantly increased in obese patients. The increase of miR-146a-5p was also accompanied by an increase of inflammatory genes, such as *TRL4, NF-kB, IL-6,* and *TNF-α* in human mononuclear leukocytes, while, after exposure to physical activity, four-fifths of responders in these obese patients had a remarkable decrease of miR-146a-5p. In addition, miR-146a-5p was significantly correlated to lipid parameters, namely total cholesterol and inflammatory cytokine *IL-8*. These results indicated that miR-146a-5p might play a critical role in obesity by activating the inflammatory response and serve as a promising biomarker for obesity and its metabolic complications during exercise.

Gilberto et al. [85] analysed the relationship between circulating miRNAs and strength training in T2DM patients and found that diabetic patients exhibited an apparent reduction of blood sugar compared to nondiabetic patients after strength training. This reduction resulted in an increase of miR-146a in the blood instead of miR-126. Evidence suggested that miR-146a^−/−^ mice exposed to a high-fat diet gained more body weight and fat mass than controls, accompanied by insulin resistance and glucose intolerance. Both adipocytes transfected with miR146a and knockout of natriuretic peptide receptor 3 (NPR3), a target gene of miR-146a, exhibited an increase in insulin-stimulated glucose intake, demonstrating that miR-146a modulated insulin sensitivity by downregulating *NPR3* [86]. These results indicated miR-146a might serve as a more promising biomarker to evaluate response to antidiabetic intervention than miR-126.

#### 4.1.2. miR-9 and miR-375

Studies showed miR-375 and miR-9 presented a positive correlation to glycaemic status. Moreover, miR-375 showed good diagnostic abilities to distinguish prediabetes and T2DM patients from healthy controls, with area under curve (AUC) of 0.76 (95% confidence interval (CI): 0.630–0.884, *p* = 0.001) and 0.77 (95% CI: 0.65–0.89, *p* < 0.001) compared to miR-9 (prediabetes: AUC = 0.63, 95% CI: 0.485–0.777, *p* = 0.08; T2DM: AUC = 0.50, 95% CI: 0.301–0.604, *p* = 0.053). Results also showed that the combination of miR-375 and miR-9 significantly enhanced the predictability to distinguish patients from controls. Taken together, miR-375 alone or combined with miR-9 could be used as biomarkers for prediabetes and T2DM [25]. Inhibiting insulin exocytosis by miR-9 and reducing the number and viability of pancreatic β-cells by overexpression of miR-375 may provide a theoretical basis for their application in T2DM diagnosis. Despite the diagnostic value or valid theoretical basis, few studies investigate the change of these two miRNAs in metabolic syndrome during exercise in cancer survivors.

#### 4.1.3. Others

Other miRNAs, such as miR-92a, miR-130a, miR-222, and miR-370, were remarkably decreased in pre-atherosclerotic patients (patients without atherosclerosis but having hypertension, hyperlipidaemia, and/or diabetes), with miR-126 and miR-130a decreased up to 50% [87]. Like miR-375 and miR-9, these miRNAs had been deeply investigated for the mechanism of the initiation of metabolic disease. Still, these studies did not establish a clinical correlation between metabolic syndrome and the miRNAs, and there was no overlap identified with physical activity or exercise training. Therefore, whether these miRNAs can be used for potential biomarkers of exercise-based cancer rehabilitation remains to be determined.

### 4.2. Cardiorespiratory Capacity

#### 4.2.1. myomiRs

The elevation of circulating miR-1 was significantly associated with LVEF reduction, and the receiver operating characteristic curve (ROC) analysis showed that miR-1 exceeded cardiac troponin I (cTnI) in distinguishing potential victims to cardiotoxicity [61]. In rat models, 14 weeks of endurance training significantly increased the expression of miR-1 and miR-133, which can result in physiological hypertrophy by targeting downstream *Srf, Hdac4,* and *Hand2* gene [88]. Moreover, circulating miR-1, miR-133a, and miR-206 were correlated to aerobic performance parameters such as VO_2_max and running speed at individual anaerobic lactate thresholds in marathon runners [89]. Nielsen et al. [90] demonstrated that the expression of myomiRs in muscle biopsies, such as miR-1, miR-133a, miR-133b, and miR-206, were related to increased VO_2_max and improved insulin sensitivity after endurance training. These results suggested that these myomiRs play a pivotal role in physical adaption during exercise. Nevertheless, Wardle et al. [69] found several myomiRs (miR-1, miR-133a, miR-206, miR-499) expressed at very low levels in the plasma. Although miRNAs at a low level are sufficient to cause physiological changes, whether or not the low level of miRNAs impacts, the correlation analysis remains unknown.

miR-208a, miR-208b, and miR-499 are all encoded by the myosin gene in muscle. Although miR-208a is cardiac-specific and its role in doxorubicin-induced cardiotoxicity has been validated in the mice model, its role in cancer patients seemed disappointing [61,91]. miR-208b and miR-499 were reported to rebound back to pre-exercise levels after 24h of completion of the exercise in marathon runners. They were not correlated with cardiac injury markers, such as cTnI and B-type natriuretic peptide (NT-pro-BNP) [89]. These results suggested the limited role of these miRNAs to evaluate or monitor cardiovascular capacity in physical activity.

#### 4.2.2. miR-126

In rat models, miR-126, as a pre-angiogenic miRNA, played a critical role in having a cardioprotective effect after exposure to crocin and voluntary exercise, the mechanism of which involved facilitated cardiac angiogenesis through *Akt* and *ERK1/2* pathways [83]. Furthermore, miR-126, which HIF-1α induced during exercise training, also contributed to myocardial angiogenesis via the *PI3K/AKT/eNOS* and *MAPK* signalling pathway and subsequently improved heart function in myocardial infarction rats [92]. Exosomes from miR-126 overexpressing mesenchymal stem cells were shown to induce angiogenesis via the *PIK3R2*-mediated *PI3K/Akt* signalling pathway [84]. Moreover, miR-126 was positively correlated to NT-pro-BNP and cTN-I in atherosclerotic patients [87]. Exercise training reduced the high-density lipoprotein (HDL)-induced miR-126 in patients with chronic heart failure and eventually caused atherogenesis and endothelial dysfunction [93]. These results suggested miR-126 was associated with common indicators of cardiac damage and may serve as a potential biomarker for treatment-induced cardiotoxicity as well we exercise-based cardiac and cancer rehabilitation.

#### 4.2.3. miR-21 and miR-146a

miR-21 and miR-146a are highly responsive to both cardiac damage and physical activity. miR-21 can attenuate diastolic dysfunction of diabetic cardiomyopathy via targeting gelsolin [94]. Like miR-126, miR-21, as a mediator, can promote hypoxia-induced angiogenesis in the cardiac microvascular endothelial cells [95]. Other studies have shown the relationship between miRNAs and common indicators of cardiac damage. miR-21 was negatively correlated to NT-pro-BNP and cTN-I among atherosclerotic patients [87]. In myocardial infarction in rat models, the injection of miR-146 antagomiR improved the reduction of left ventricular ejection fraction LVEF and fractional shortening. miR-146 antagomiR also reduced the levels of atrial natriuretic peptide (ANP) and BNP mRNA. In addition, miR-146 myomiRs attenuated cardiac fibrosis by reducing the increase of collagen I and collagen III mRNA. These results demonstrated that inhibition of miR-146 ameliorated cardiac dysfunction and remodelling [96]. A linear correlation between circulating miR-146a and cardiac marker creatine kinase-MB (CKMB) isoenzyme as well as high sensitivity C-reactive protein (hs-CRP) was observed in basketball players [97]. Studies showed that miR-21 and miR-146a had an opposite expression pattern and function during exercise: miR-21 had proinflammatory effects and was upregulated after acute exercise but downregulated after endurance training. By contrast, miR-146a had a proinflammatory effect and was upregulated by acute cycling exercise before and after 90 days of sustained rowing training. Additionally, the expression of miR-21 was negatively correlated with VO_2_max, while miR-146a was positively correlated with VO_2_max [98,99], suggesting promising prospects in an exercise-based rehabilitation program.

#### 4.2.4. miR-222

In a study with *HER-2* positive breast cancer patients undergoing neoadjuvant target therapy Trastuzumab, serum miR-222-3p protected against the relative drop of LVEF (OR = 0.410, 95% CI: 0.175–0.962, *p* = 0.040) and absolute drop of LVEF (OR = 0.394, 95% CI: 0.166–0.937, *p* = 0.035), demonstrating that miR-222-3p was an independent protective factor for trastuzumab-induced cardiotoxicity [100]. Recent work suggested that high-intensity interval training increased miR-222 expression after an acute bout of exercise and following a sustained period of exercise [101]. Liu et al. [102] suggested that miR-222 is necessary for exercise-induced cardiac growth and to counteract pathological cardiac remodelling via targeting *P27, HIPK1,* and *HMBOX1*. This may explain how exercise benefits cardiovascular fitness. Despite the positive effects of miR-222 on cardiac physiology, miR-222 can inhibit *p27^Kip1^*, a cyclin-dependent kinase inhibitor, thus promoting cancer growth [103]. Moreover, a low level of serum miR-222 was associated with superior pathological complete response, disease-free survival, and overall survival in breast cancer patients who received Trastuzumab [100]. These results suggested that miR-222 was a promising biomarker for cardiac fitness but a double-edged sword in the QoL of cancer survivors.

#### 4.2.5. miR-155

In fibrotic illnesses, miR-155 is persistently elevated, and knocking it out reduces collagen synthesis [104]. In vivo studies showed that miR-155-5p was significantly downregulated in radiation-induced EMT cell model, while in vitro studies showed that ectopic overexpression of miRNA-155-5p inhibited EMT through the NF-KB pathway by targeting *GSK-3β*. This alleviated radiation-induced pulmonary fibrosis [105]. In the mouse model of bleomycin-induced pulmonary fibrosis, the *miRNA–155/LXR* pathway was identified to play an important role in experimental and idiopathic pulmonary fibrosis [106]. In another study, untrained rats showed enhanced acetylcholine (Ach)-induced relaxation and nitric oxide (NO) production compared to trained rats. Results showed a higher expression of endothelial NO synthase (eNOS), the posttranslational modification of which by the *PI3-kinase/Akt1/2/3* pathway, contributed to higher production of exercise-induced NO expression, which was associated with a lower level of miR-155, indicating that the favourable effects of exercise on the circulating system may involve the participation of miR-155 in the molecular mechanism of improved vascular relaxation in muscular arteries [107]. Despite an increase of miR-155 in blood and no relationship between miR-155 and VO_2_max in healthy people after exercise training, the relationship between miR-155 and respiratory fitness in treatment-induced pulmonary patients remains to be determined [89,108].

#### 4.2.6. Others

A systematic review for breast cancer patients suggested that let-7, miR-20a, and miR-210 may serve as biomarkers for anthracycline-based cardiotoxicity during chemotherapy [58]. In the rat model treated with doxorubicin at increasing doses, the level of cTnT was significantly elevated in the 18mg/kg dose. The decrease of let-7g levels accompanied it. Only a few studies investigated the change of let-7 in cardiotoxic patients after physical activity [109]. By targeting *PTEN*, miR-20a prevented exercise-associated CVD, but whether miR-20a played a role in reversing treatment-induced CVD remains unknown [110]. Accumulating evidence suggested that the expression of miR-210 was inversely associated with VO_2_max after physical activity. A possible explanation is that, during exercise, a hypoxic condition in cardiac muscle triggered angiogenesis through *HIF-1α*-induced miR-210 [83]. Further efforts are required to establish a correlation between these miRNAs and physical and physiologic parameters in cancer survivors.

### 4.3. Cancer Cachexia

#### 4.3.1. miR-21

Evidence suggested that miR-21 was upregulated in denervated muscles. miR-21, along with miR-206, was sufficient for the atrophy process by targeting *YY1* and *eIF4E3* [111]. He et al. [112] showed that tumour-derived microvesicles containing miR-21 could induce apoptosis of muscle cells through Toll-like 7 receptors on murine myoblasts. Moreover, c-Jun N-terminal kinase activity is required to mediate this apoptotic response. This may shed light on how tumour cells promote loss of muscle mass. Taken together, miR-21 plays an important role in myogenesis and muscle mass gain. Other studies showed that low psoas muscle mass index was a prognostic biomarker for metastasis in colorectal cancer patients. The level of serum miR-21 was significantly elevated in colorectal cancer patients with low psoas muscle mass. This indicated that serum miR-21 quantification might help clinicians make decisions on developing intervention strategies for these patients [113]. However, it should be noted that miR-21 also regulated muscle loss in a natural process, such as ageing [114]. Additionally, evidence showed that levels of serum miR-21 and other miRNAs, such as miR-146a, miR-221, and miR-222, are unrelated to muscle bulk and fat reserve in athletes [97]. Few studies were conducted to explore the relationship between miR-21 and muscle mass and strength gain induced by physical activity, which therefore require further investigation.

#### 4.3.2. miR-378

In vivo study showed that training-induced change of miR-378 was positively associated with muscle mass gains. However, miR-378 was downregulated in low responders to resistance exercise amongst weightlifters. In contrast, an increase in miR-378 was not observed in high responders [39]. Aside from the loss of muscle mass, cancer cachexia is also associated with a significant loss of adipose tissue, which is caused by enhanced lipolysis in adipocytes in some but not all patients. Kulyte et al. [64] detected the expression of miR-378 in abdominal subcutaneous adipose tissue from gastrointestinal cancer patients and found that miR-378 was significantly elevated in cachectic cancer patients compared to non-cachectic patients. Overexpression of miR-378 was positively associated with loss of adipose tissue, which involved catecholamine-stimulated lipolysis in adipocytes. Inversely, inhibition of miR-378 can reduce the expression of *LIPE, PLIN1,* and *PNPLA2* genes, which encoded key regulators for lipolysis and therefore decreased catecholamine-stimulated lipolysis. In vivo study showed that exercise-induced change in miR-378 abundance was positively correlated to muscle mass gain [39].

#### 4.3.3. miR-1 and miR-133

Although miR-1 and miR-133 are clustered on the same chromosomal loci, they have very distinct roles in transcriptional circuits and therefore differentially regulate skeletal muscle cell proliferation and differentiation. miR-1 promotes myogenesis by targeting transcriptional gene histone deacetylase 4 in muscle, while miR-133 enhances myoblast proliferation by inhibiting serum response factor [115]. Moreover, miR-1 and miR-133 show different expression patterns in response to aerobic and resistance exercises. Both miRNAs were upregulated within three hours following an acute aerobic exercise bout but downregulated after four weeks of aerobic endurance training [116]. Again they decreased after 6 h of resistance training and remained downregulated after a week of resistance exercise training [116]. Recent work suggested that reduced IGF-1 protein was associated with upregulated miR-133 during the differentiation of C2C12 cells. Overexpression of miR-133 during differentiation significantly suppressed the expression of the IGF-1 receptor at the transcriptional level by downregulating the phosphorylation of *Akt*. Moreover, the upregulation of miR-133 can be accelerated by the addition of IGF-1. These results indicated that metabolism-related factors played an important role in myogenesis [116,117]. MiR-133a-deficient mice had a low maximal exercise capacity and low resting metabolic rate. The downregulated transcription of a set of mitochondrial biogenesis regulators, such as nuclear respiratory factor-1, transcription factor A, and so on, may explain lower mitochondrial mass and impaired exercise capacity in miR-133a-deficient mice. Six weeks of endurance training improved the expression of miR-133a and stimulated mitochondrial biogenesis in wild-type mice but failed to enhance the mitochondrial function in miR-133a-deficient mice [118]. Mechanistic analysis showed IGF-1 receptor, the target of miR-133a, and the hyperactivation of *Akt* signalling were involved in the lower transcription of the mitochondrial biogenesis regulators [118]. These results suggested that miR-133a reduced exercise capacity by lowering mitochondrial function. However, few studies were conducted to establish the correlation between the expression of miR-1 and miR-133 and muscle mass or strength after exercise training.

### 4.4. Depression and Anxiety

#### 4.4.1. let-7

Evidence showed that depressed patients with 3-month treatment with antidepressants reported an elevation of a series of let-7 family members in blood, suggesting let-7 may be a therapeutic target in depression [119]. In a genetic rat model of depression, the decreased let-7 could increase the levels of the proinflammatory cytokine *IL-6*, leading to depression. On the other hand, running can reduce the level of *IL-6* and rescue the expression of let-7i. This can elevate mood, thus exerting an antidepressant-like effect [120]. Studies showed that the baseline level of let-7b in treatment-resistant depression patients was 40% lower than healthy controls. In comparison, let-7c was 50% lower in treatment-resistant depression patients who received electroconvulsive therapy when compared to healthy controls. Their role in depression may involve in *PI3k-Akt-mTOR* signalling pathway [121]. Hung et al. [122] demonstrated that the level of let-7e before starting antidepressant treatment was inversely correlated to the severity of depression. The expression of let-7e was also altered by antidepressant treatment, but whether let-7e levels were affected by exercise training remains unknown.

#### 4.4.2. miR-132 and miR-182

In vitro study showed that miR-132 and miR-182 reduced the expression of brain-derived neurotrophic factor (BDNF), which was greatly important in the aetiology of depression. Patients with depressive symptoms had lower BDNF levels and higher miR-132 and miR-182 levels in blood compared to healthy controls. This supports the role of miR-132 and miR-182 in regulating BDNF expression [123]. Additionally, the self-rating depression scale (SDS) score was negatively correlated to serum BDNF levels but positively correlated to serum miR-132 levels. In contrast, the serum BDNF expression was reversely associated with the expression of miR-132/miR-182 [123]. In a rat model, downregulation of miR-132 can mitigate behavioural impairment of rats exposed to single prolonged stress, the mechanism of which may involve methyl GpG-binding protein 2, a positive mediator of BDNF regulated by miR-132-3p [124]. These results indicated that miR-132 and miR-182 may play a critical role in the BDNF-mediated pathway and may serve as diagnostic or therapeutic targets of mental disorders. Although a change of miR-132 was observed after endurance training, which can also reduce depressive symptoms in humans, whether the positive effect of exercise on depression involved the miR-132 regulatory pathway remains to be determined.

#### 4.4.3. miR-134

An enriched environment could ameliorate depressive-like behaviours caused by chronic unpredictable mild distress (CUMS). The *SIRT1/miR-134* signalling pathway regulates its downstream molecules, including synaptic plasticity proteins and BDNF expression in primary cultured hippocampal neurons. The protective effect of an enriched environment may be through activating the *SIRT1/miR-134* signalling pathway, remodelling the dendritic spine, altering synaptic ultrastructure, and increasing synaptic plasticity proteins and BDNF expression in the hippocampus [14].

#### 4.4.4. miR-34b/c

After exposure to NBI-27914, a specific corticotropin-releasing hormone receptor 1 (CRHR1) antagonist, which was correlated to trauma-induced anxiety, the rat model was observed for changes in the hypothalamic-pituitary adrenal (HPA) axis and anxiety-like behaviour. Bioinformatic analysis showed that CRHR1 was the target of miR-34b, and the overexpression of miR-34b negatively modulated CRHR1 mRNA in the primary hypothalamic neurons, mitigating the hyperactivity of the HPA axis and anxiety-like behaviour [125]. Evidence also showed that miR-34b/c had a negative impact on the cognitive function of major depressive patients [126]. However, few studies explored how physical exercise alleviates psychological symptoms by modulating the expression of miR-34b/c.

#### 4.4.5. Others

In addition to miR-132 and miR-134, CUMS gives rise to depression-like behaviours via *SIRT1/miR-124*, and swimming exercise can reverse the expression of SIRT1 protein and the expression of these miRNAs in CUMS mice. However, the correlation between depressive behaviour and epigenetic changes of hippocampal plasticity was not established [81]. In major depressive disorder patients, intracellular miR-146a level was negatively correlated with the severity of depression; by contrast, miR-155 showed the opposite effect [122]. However, their role in exercise-related prevention of depression requires further investigation.

## 5. The Challenge of miRNAs as Biomarkers in Exercise-based Cancer Rehabilitation

Albeit the promising prospect of miRNAs as potential biomarkers for exercise-based cancer rehabilitation, there are several issues remaining to be resolved: (1) clinical correlation between miRNAs expression in specific tissues and their expression in blood as well as physiological and physical parameters in cancer rehabilitation should be established due to the lack of related investigation; (2) the changing pattern of miRNAs differs from conventional biomarkers after exercise: NT-proBNP and hs-CRP increased after the marathon and stay elevated after 24h of race completion. By contrast, circulating miRNAs elevated instantly after the race and returned to prerace level or even lower 24h postrace. These results suggested that circulating release and clearance mechanisms of miRNAs may differ from traditional biomarkers and that miRNAs may serve as real-time and simultaneous instead of long-term markers of exercise-induced muscle adaptation [127]. In addition, most studies investigated the effect of physical activity on miRNAs among healthy people or athletes, and few studies focus on the change of miRNAs on the deconditioned cancer patients during exercise. Therefore, it is necessary to conduct research exploring the effect of exercise-based rehabilitation on the dynamic change of miRNAs in cancer survivors; (3) miRNAs were associated with age, gender, training modality, or exercise regimen; we should, therefore, adjust these factors before we put miRNAs in use; (4) miRNAs at very low-level change are sufficient to cause an imbalance of human body by binding to multiple target genes [69]. Thus, the target gene in a regulatory network needs to be fully elucidated. Thus, the target gene, as an auxiliary tool for miRNAs in rehabilitation, opens a new opportunity for future investigation.

## Figures and Tables

**Figure 1 life-11-01439-f001:**
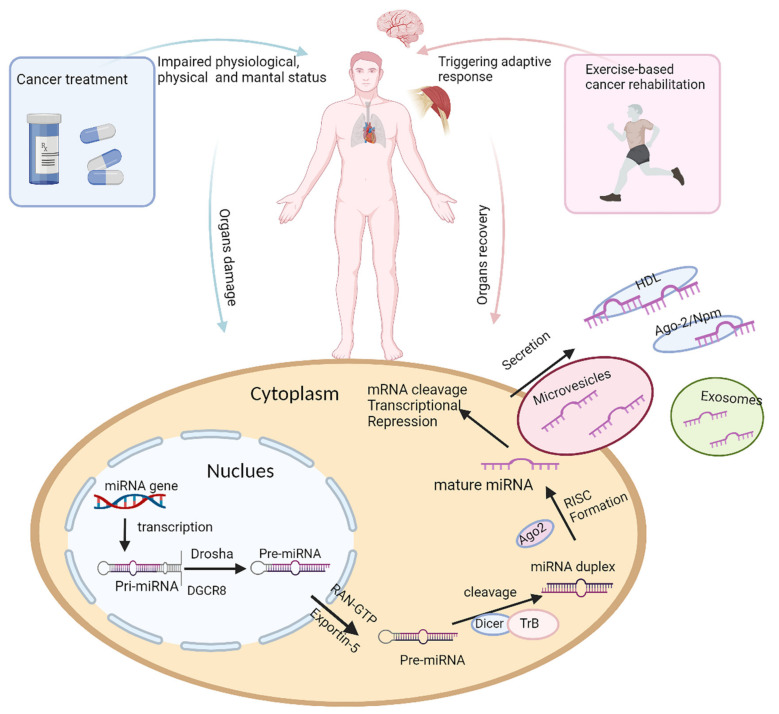
The impact of cancer treatment and physical activity on miRNA biogenesis. Organ damage can be caused as a result of cancer or as an adverse effect of cancer treatment, leading to decreased physical and mental ability and poor quality of life for cancer survivors, while exercise-based cancer rehabilitation triggers adaptive responses, resulting in altered miRNA expression, in damaged organs, thus promoting in organ recovery and improved quality of life. The diagram illustrates the changes in miRNA expression that can occur during tissue damage, toxicity, treatment side effects, and during physical activity, and this is part of the adaptive process or physiologic responses to exercise. During tissue damage, miRNAs may be released into circulation. miRNA biogenesis and molecular mechanisms of miRNA production can also be altered by disease or as a response to physical activity. miRNA biogenesis begins in the nucleus, where primary miRNA (pri-miRNA), transcribed by miRNA gene, is processed by Drosha and Dgcr8 into stem-looped structures, precursor miRNA (pre-miRNA). Pre-miRNA is exported by Exportin 5 into cytoplasm, where mature miRNA is produced after further processing by Dicer. Mature miRNA is transported out of cell by either exocytosis or combining protein complex, regulating gene expression in cancer treatment-induced complications or adaptive changes during exercise-based cancer rehabilitation.

**Figure 2 life-11-01439-f002:**
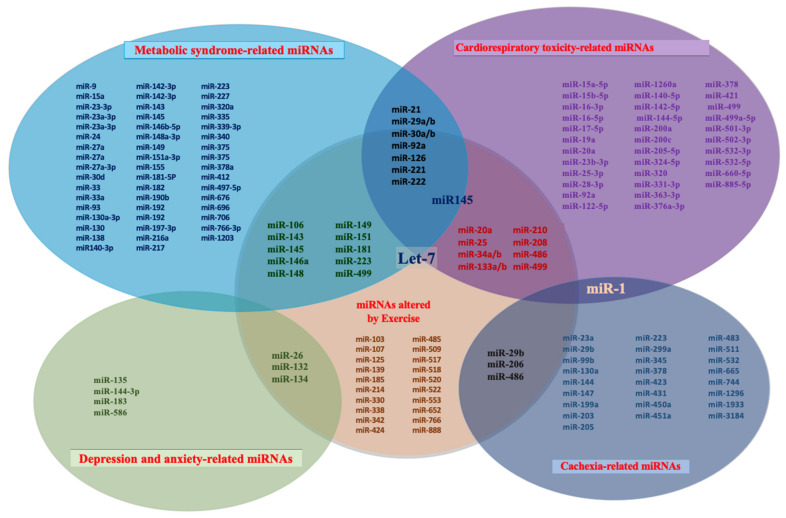
MiRNA expression in cancer treatment-induced complications and physical activity. The Venn diagram shows the differentially expressed miRNA in both common cancer treatment-induced complications (metabolic syndrome, cardiorespiratory toxicity, cachexia, depression, and anxiety) and physical activity (endurance training, aerobic training, or both) from literature. The overlapping miRNA may be potential biomarkers for exercise-based cancer rehabilitation for cancer survivors but remains to be investigated.

**Table 1 life-11-01439-t001:** The application of miRNAs in diagnosis, treatment, and prognosis in cancer.

AntagomiRs					
No.	Disease	Targeted miRNAs	Sample	Findings and Mechanisms	Ref.
1	NPC	EBV-miR-BART10-5p, Hsa-miR-18a	In vitro In vivo	Inhabitation of angiogenesis Overexpression of EBV-miR-BART10-5p and hsa-miR-18a promotes angiogenesis in vitro and in vivo by regulating the expression of VEGF and HIF1-a in a Spry3 (tumour suppressor)-dependent manner	[27]
2	BC	miR-155	Cell line	Inhabitation of proliferation and promotion of apoptosis by increasing the expression of TP53INP1	[28]
3	CRC	miR-21	Cell line	Increase of posttranscriptional gene slicing protein and mRNA Downregulation of angiogenesis-associated miR-30	[29]
4	BC	miR-10b	Mice	Increased expression of HOXD10 leading to decreased expression of pro-metastatic gene RHOC	[30]
Diagnostic and prognostic biomarkers					
No.	Disease, study design, population	miRNAs	Sample	Results	Ref.
1	Cancer patients * (*n* = 163) vs. healthy controls (*n* = 63)	miR-195, let-7a, miR-10b, miR-155	Whole blood sample	Let 7a, miR-10b, and miR-155 expressed differentially but non-specifically in majority of cancer patients; circulating miR-195 was breast cancer-specific miR-195 differentiated breast cancer patients from other cancers and from healthy controls with a sensitivity of 88% at a specificity of 91%	[31]
2	BC, case-control, BC (*n* = 170) vs. healthy controls (*n* = 100) vs. other types of cancer (*n* = 95);Validation cohort: BC (*n* = 70) vs. healthy controls (*n* = 50)	miR-145, miR-451	Plasma, cancer tissue	3 miRNAs significantly increased before surgery and reduced after surgery in BC patients A combination of miR-145 and miR-451 yielded ROC of 0.931 in discriminating BC from healthy controls and other cancers, with a PPV of 88% and NPV of 92%	[32]
3	EEC, EEC (*n* = 77) vs. controls (*n* = 45)	A set of miRNAs	Plasma, cancer tissue	miR-92a/miR-410 (AUC: 0.977) and miR-92a/miR-205/miR-410 (AUC: 0.984) differentiated tumour tissues with higher accuracy Tissue miR-205/miR-200a predicted relapse with AUC of 0.854 Tissue miRNA signatures were independent prognostic markers of overall (miR-1228/miR-200c/miR-429, HR: 2.98) and progression-free survival (miR-1228/miR-429, HR: 2.453) Plasma miR-9/miR-1228 (AUC: 0.909) and miR-9/miR-92a (AUC: 0.913) differentiated EEC with higher accuracy	[33]
4	GC, GC (*n* = 123) vs. healthy controls (*n* = 111)	miR-627, miR-629, and miR-652	plasma	miR-627, miR-629, and miR-652 were significantly higher in gastric cancer patients than healthy controls A combination of 3 miRNAs obtained the highest AUC of 0.942, with a cut-off at 0.373, with a sensitivity of 86.7% and a specificity of 85.5%	[34]

NPC: nasopharyngeal carcinoma; BC: breast cancer; CRC: colorectal cancer; EBV: Epstein–Barr virus; VEGF: vascular endothelial growth factor; HIF: hypoxia inducible factor; HOXD10: homeobox D10; cancer patients *: including breast, prostate, colon, and renal cancer and melanoma; NSCLCs: non-small cell lung carcinoma; EEC: endometrioid endometrial cancer; AUC: area under curve; GC: gastric cancer.

## Data Availability

Not applicable.

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
