# Peer review of "MicroRNAs as Potential Biomarkers for Exercise-Based Cancer Rehabilitation in Cancer Survivors"

_life, 2021, doi:10.3390/life11121439_

Round 1

Reviewer 1 Report

The manuscript is interesting and clear

Author Response

Life-1482441 Responses to reviewers

Reviewer 1:

Comment 1. The manuscript is interesting and clear.

Responses: Thanks very much for your appreciation. Thank you for taking time to review our manuscript

Reviewer 2 Report

life-1482441

General Comments

This manuscript

This manuscript describes significance of miRNA in cancer patients. Although the focus on non-neoplastic and/or indirect molecular alternation is interesting, this manuscript needs further elaboration before publication.

Major Concerns

Figure 1. Significance of exercise remains unclear. Is there any effect on molecular process of miRNA? Figure 1 includes the gap between organ (also called as macroscopic?) and molecular levels.

In the main text, some abbreviations are defined twice.

Genes should be written in italic.

Line 157. “P13K” seems to be PI3K.

Figure 2. The image seems to be busy due to abundant miRNA. It may be confusing. In addition, some miRNAs, for example, let-7, appear at two and more location. Is it possible to unify the duplicated ones into the single ones in the Venn diagram?

Main text tends to be redundant. To enhance the readability of the manuscript, please shorten the text or summarize as a table.

Generated reference seems to be inconsistent and lacks some bibliographic information. Please carefully check and correct it.

Supplementary table should be moved into manuscript if possible.

Author Response

Life-1482441 Responses to reviewers

Reviewer 2:

Comment 1.  Figure 1. Significance of exercise remains unclear. Is there any effect on molecular process of miRNA? Figure 1 includes the gap between organ (also called as macroscopic?) and molecular levels.

Response: Thanks for your comments. According to reviewer’s comment, we have revised Figure 1. The diagram illustrates the changes in miRNA expression that can occur during tissue damage, toxicity, treatment side effects as well as during physical activity and this is part pf the adaptive process or physiologic responses to exercise. During tissue damage miRNAs may be released into circulation. miRNA expression is also altered during physical activity. miRNA biogenesis and molecular mechanisms of miRNA production can also be altered by disease or as a response to physical activity. 

We have rewritten the legends to make support the figure more clearly:

   Figure 1. The impact of cancer treatment and physical activity on miRNA biogenesis  Organ damage can be caused as a result of cancer or as a adverse effect of cancer treatment leading to decreased physical and mental ability, and poor quality of life for cancer survivors; while exercise-based cancer re-habilitation triggers adaptive responses, resulting in altered miRNA expression, in damaged organs, thus promoting in organ recovery and improved quality of life. The diagram illustrates the changes in miRNA expression that can occur during tissue damage, toxicity, treatment side effects as well as during physical activity and this is part pf the adaptive process or physiologic responses to exercise. During tissue damage miRNAs may be released into circulation. miRNA expression is also altered during physical activity. miRNA biogenesis and molecular mechanisms of miRNA production can also be altered by disease or as a response to physical activity. miRNA biogenesis begins in the nucleus, where primary miRNA (pri-miRNA), transcribed by miRNA gene, is processed by Drosha and Dgcr8 into stem-looped structures, precursor miRNA (pre-miRNA). Pre-miRNA is exported by Exportin 5 into cytoplasm, where mature miRNA is produced after the further process by Dicer. Mature miRNA is transported out of cell by either exocytosis or combining protein complex, regulating gene expression in cancer treatment-induced complications or adaptive changes during exercise-based cancer rehabilitation.

In the main text, some abbreviations are defined twice.

Response: Thanks for your comments. According to reviewer’s comment, we have corrected the mistakes.

Genes should be written in italic.

Response: Thanks for your comments. According to reviewer’s comment, genes in the manuscript have been corrected in italic style.

Line 157. “P13K” seems to be PI3K.

Response: Thanks for your comments. According to reviewer’s comment, we have corrected it as PI3K.

Figure 2. The image seems to be busy due to abundant miRNA. It may be confusing. In addition, some miRNAs, for example, let-7, appear at two and more location. Is it possible to unify the duplicated ones into the single ones in the Venn diagram?

Response: We have edited the figure to make it more clear and removed any duplicates. The common miRNAs are shown in the overlapping regions of the Venn diagram in bold.

Main text tends to be redundant. To enhance the readability of the manuscript, please shorten the text or summarise as a table.

Response: Thanks for your comments. According to reviewer’s comment, the main text has been carefully refined and any duplicates or redundant text found was removed.

Generated reference seems to be inconsistent and lacks some bibliographic information. Please carefully check and correct it.

Response: Thanks for your comments. According to reviewer’s comment, the references have been carefully checked and corrected.

Supplementary table should be moved into manuscript if possible.

Response: Thanks for your comments. According to reviewer’s comment, the supplementary table has been moved to manuscript, as shown in Table 1.

Reviewer 3 Report

The authors provided a well-documented overview of microRNAs role and application in cancer. In addition to this, the review could represent a really interesting point of view in a field so dynamic and rich in potential future applications of miRNAs as nanotherapeutics. The field of research focused on circuloma is in continuous evolution and even if the article is well written, the introduction section could be improved with a more general point of view about the application of exosome associated mirnas  in other fields of research adding some recent works related to the importance of exosomes in other diseases (PMID: 32932746 is just an example). 

I hope that my comments could be useful for a better version of the paper.

Good luck!

Author Response

Life-1482441 Responses to reviewers

Reviewer 3:

Comment 1. The authors provided a well-documented overview of microRNAs role and application in cancer. In addition to this, the review could represent a really interesting point of view in a field so dynamic and rich in potential future applications of miRNAs as nanotherapeutics. The field of research focused on circuloma is in continuous evolution and even if the article is well written, the introduction section could be improved with a more general point of view about the application of exo-some associated mirnas in other fields of research adding some recent works related to the importance of exosomes in other diseases (PMID: 32932746 is just an example).

Response: Thanks for your comments. According to reviewer’s comment, we have added a paragraph on the introduction section including some recent works on the application of exosome derived miRNAs in other fields.

Round 2

Reviewer 2 Report

The authors responded the comments of the reviewer. Although the revised manuscript have been improved, Figure 2 have some fonts which are hard to see in Black and White printing. Please confirm it. 

Author Response

We have updated Figure 2 to make the fonts bigger and more visible in bold format. 
